# Exploring the molecular composition of the multipass translocon in its native membrane environment

Max Gemmer, Marten L Chaillet ⬤, Friedrich Förster ⬤

**Multispanning membrane proteins are inserted into the endoplasmic reticulum membrane by the ribosome-bound multipass translocon (MPT) machinery. Based on cryo-electron tomography and extensive subtomogram analysis, we reveal the composition and arrangement of ribosome-bound MPT components in their native membrane environment. The intramembrane chaperone complex PAT and the translocon-associated protein (TRAP) complex associate substoichiometrically with the MPT in a translation-dependent manner. Although PAT is preferentially part of MPTs bound to translating ribosomes, the abundance of TRAP is highest in MPTs associated with non-translating ribosomes. The subtomogram average of the TRAP-containing MPT reveals intermolecular contacts between the luminal domains of TRAP and an unknown subunit of the back-of-Sec61 complex. AlphaFold modeling suggests this protein is nodal modulator, bridging the luminal domains of nicalin and TRAPα. Collectively, our results visualize the variability of MPT factors in the native membrane environment dependent on the translational activity of the bound ribosome.**

## Introduction

The ribosome-associated endoplasmic reticulum (ER) translocon complex facilitates the biogenesis of most secretory and membrane proteins (Gemmer & Forster, 2020). The ribosome binds to the protein-conducting channel Sec61, a heterotrimeric ER membrane-embedded complex, which facilitates signal peptide (SP) insertion into its lateral gate and co-translational translocation of the nascent chain into the ER lumen through its central pore (Park & Rapoport, 2012). To meet the requirements of the broad spectrum of nascent protein clients, Sec61 associates with various accessory factors specialized in functions such as SP insertion and cleavage, N-glycosylation, protein folding and maturation, transmembrane helix insertion, and ER stress response (Lang et al, 2017; Bai & Li, 2019; Gemmer & Forster, 2020; Liaci & Forster, 2021; Hegde & Keenan, 2022). In all mammalian cells studied to date, the most abundant

ribosome-bound ER translocon variant comprises the translocon-associated protein (TRAP) complex and the oligosaccharyltransferase A (OSTA) complex in addition to Sec61 (Pfeffer et al, 2015). A less abundant variant comprises only Sec61 and the TRAP complex. The heterotetrameric TRAP facilitates the insertion of SPs with below-average hydrophobicity and above-average glycine-and-proline content (Fons et al, 2003; Nguyen et al, 2018), whereas OSTA mediates co-translational N-glycosylation and recruitment of ER luminal chaperones (Schwarz & Aebi, 2011; Wild et al, 2018).

In contrast to secretory proteins, biogenesis of multispanning transmembrane proteins relies on different translocon variants that specialize in transmembrane helix (TMH) insertion, membrane protein topogenesis, folding, and assembly (Hegde & Keenan, 2022). Recently, Get1, EMC3, and TMCO1 have been identified as ER-resident Oxa1 superfamily members, which are core components of different insertase complexes (Anghel et al, 2017). The guided entry of tail-anchored protein (GET) complex facilitates post-translational targeting and insertion of tail-anchored membrane proteins (Stefanovic & Hegde, 2007; Mariappan et al, 2011; Wang et al, 2011, 2014), whereas the ER membrane complex (EMC) and a complex containing TMCO1 facilitate co-translational TMH insertion of multispanning membrane proteins (Anghel et al, 2017; Chitwood et al, 2018; Guna et al, 2018; Shurtleff et al, 2018; McGilvray et al, 2020; Pleiner et al, 2020). Biochemical, mass-spectrometry, and cryo-electron microscopy (cryo-EM) studies of affinity-tagged TMCO1 isolates revealed that TMCO1 is part of a ribosome-bound translocon that is distinct from its OSTA-containing counterpart and engages with multipass-transmembrane substrates (McGilvray et al, 2020). More detailed analysis using specific substrates detailed the components and function of what is herein referred to as the multipass translocon (MPT) (Smalinskaitė et al, 2022; Sundaram et al, 2022).

Besides Sec61, the MPT comprises three sub-complexes: the GET- and EMC-like (GEL) complex, the protein associated with the ER translocon (PAT), and the back-of-Sec61 (BOS) complexes. The GEL complex consists of TMCO1 and obligate partner of TMCO1 insertase (OPTI) and it facilitates TMH insertion (Lewis & Hegde, 2021; Smalinskaitė et al, 2022; Sundaram et al, 2022). The PAT complex consists of the intramembrane chaperone Asterix to protect TMHs with exposed hydrophilic residues (Chitwood & Hegde, 2020), and

Structural Biochemistry, Bijvoet Center for Biomolecular Research, Utrecht University, Utrecht, Netherlands

Correspondence: f.g.forster@uu.nl

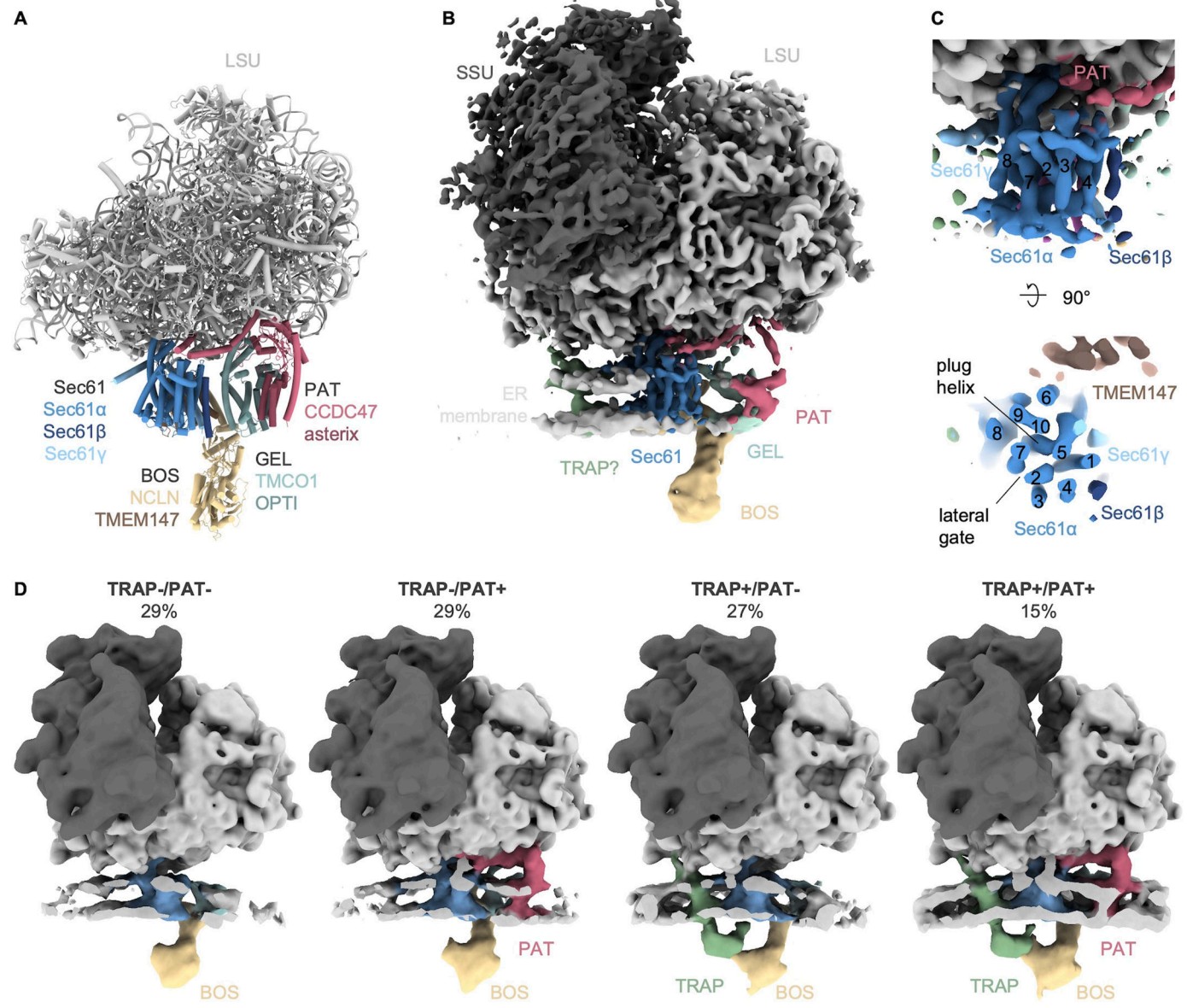

**Figure 1.   Structure and composition of the multipass translocon (MPT) in the native ER membrane.**
**(A)** Structure of the ribosome-associated MPT (PDB-7TUT) determined by Smalinskaitė et al (2022). **(B)** Cryo-ET reconstruction of the heterogenous ribosome-associated MPT-population (14,671 particles). Ribosome and Sec61 were filtered to 8 Å and the accessory factors and ER membrane area to 20 Å. **(B, C)** Close-up front (top panel) and top (bottom panel) view of Sec61 from the MPT population from (B). Numbering of transmembrane helixs of Sec61α is indicated. **(D)** Reconstructions of different MPT populations. Reconstructions were filtered to 20 Å resolution. The ER membrane was clipped for visual clarity.

CCDC47, which forms contacts with the ribosome and impedes Sec61 opening (Smalinskaitė et al, 2022). The BOS complex comprises TMEM147, nicalin (NCLN), and one of the three nearly identical paralogs of nodal modulator (NOMO) 1, NOMO2, or NOMO3, collectively referred to as NOMO (Haffner et al, 2004, 2007; Dettmer et al, 2010). The function of the BOS complex is poorly understood. TMEM147, as well as *Caenorhabditis elegans* homologs of NCLN (NRA-2) and NOMO (NRA-4), affect levels and subcellular localization of different multi-spanning membrane proteins, whereas NOMO has been shown to play an additional role as ER sheet shaping protein (Almedom et al, 2009; Kamat et al, 2014;

Christodoulou et al, 2020). Together, the components of the MPT form a lipid-filled cavity adjacent to Sec61, which mediates TMH insertion and folding of multi-spanning membrane proteins (Smalinskaitė et al, 2022; Sundaram et al, 2022). Although biochemical analysis and cryo-EM studies revealed the organization of the detergent-solubilized MPT bound to substrates, its compositional variation in the native membrane context and the correlation with ribosomal activity remains to be explored.

Here we analyze the composition and organization of the ribosome-bound MPT in the native ER membrane by re-visiting electron cryo tomograms of ER microsomes from Gemmer et al

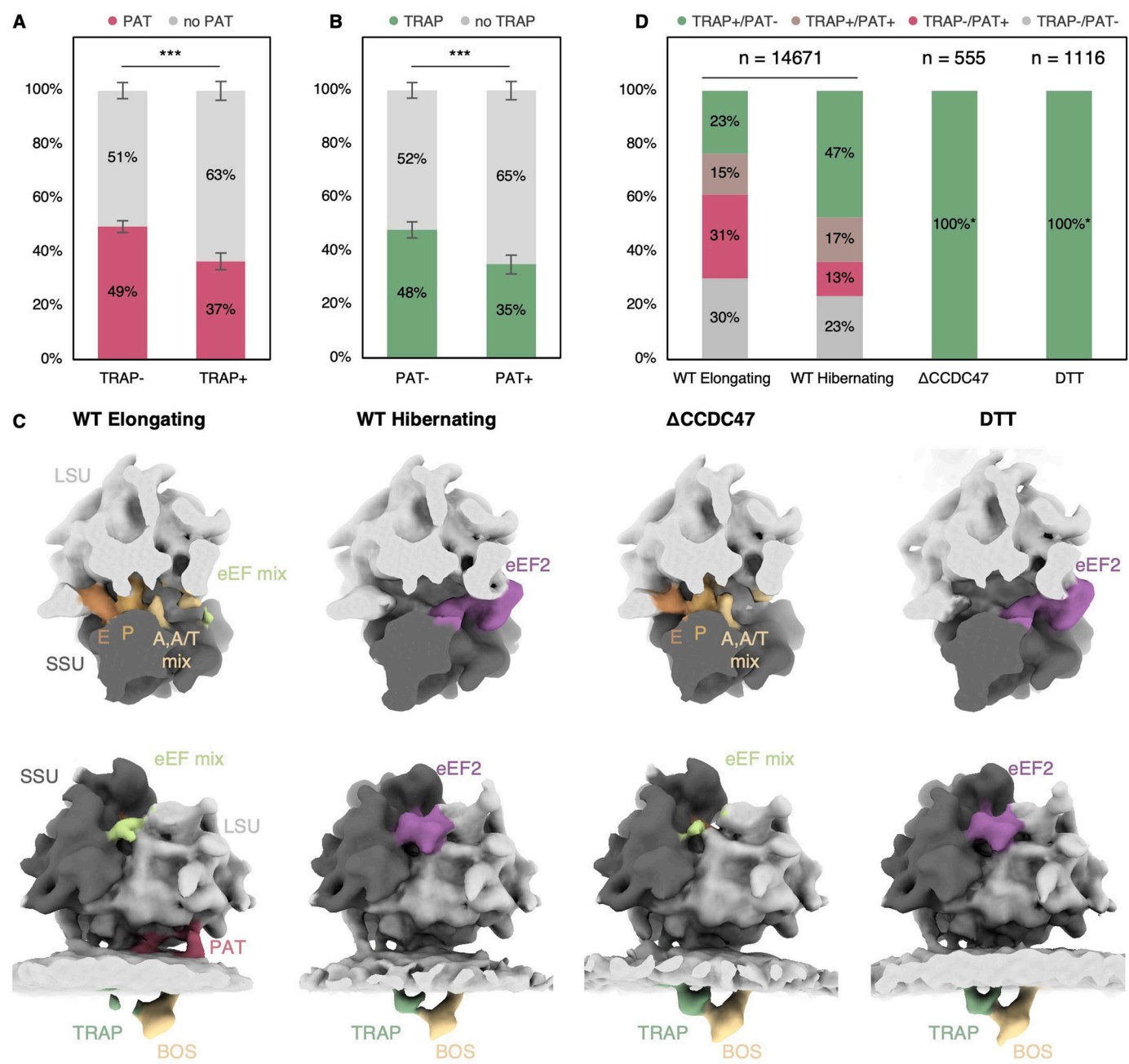

**Figure 2. Recruitment behavior of PAT and translocon-associated protein (TRAP).**
**(A)** Relative abundance of PAT (magenta) in the TRAP-lacking (TRAP−) or -containing (TRAP+) multipass translocon (MPT) population resulting from subtomogram classification of ribosome–translocon complexes on ER-derived vesicles. The distributions differ significantly with $P = 7.14 \times 10^{-12}$. **(B)** Relative abundance of TRAP (green) in the PAT-lacking (PAT−) or PAT-containing (PAT+) MPT subtomogram-average population with $P = 8.52 \times 10^{-11}$. **(C)** Reconstructions of MPT-associated ribosomes obtained from HEK293F under unperturbed conditions (WT) of elongating and hibernating classes, ΔCCDC47, and DTT-treated samples. Top row: top views of the ribosomal binding cleft and elongation factor binding site. The ribosome was clipped for clarity. Bottom row: corresponding front views of the ribosome-associated translocon populations. The ER membrane was again clipped for clarity. The subtomogram averages represent the entire multipass population in each class (hibernating and elongating) or sample (ΔCCDC47 and DTT). Note the varying occupancy of TRAP and PAT in different classes. **(C, D)** Distribution of TRAP and PAT in the multipass populations from (C) color-coded as indicated. (*) PAT-containing or TRAP-lacking classes were not detected in ΔCCDC47 and DTT-treated samples.

(2023). Whereas earlier analysis of these data covered distinct ribosomal intermediate states, major translocon variants, and the Sec61-TRAP-OSTA translocon, we now focus on subtomogram analysis of ribosome–MPT complexes. The analysis reveals the compositional differences of the MPT dependent on translation activity and ER stress.

## Results and Discussion

We previously analyzed ~135,000 ribosome particles from HEK293 cell-derived ER microsomes and revealed the distribution of the major ER translocon variants (Fig S1). We have shown that ~14,700 ribosomes were associated with the MPT variant (Fig 1A and B). Although refinement of the ribosome-bound MPT yielded a reconstruction with a resolution of ~8 Å in the vicinity of Sec61, its associated accessory factors remained poorly resolved (Fig 1B).

The Sec61 lateral gate and plug helix both adopt a closed conformation that matches its structure in the solubilized, isolated ribosome MPT (McGilvray et al, 2020) (Figs 1C and S2A and B). This contrasts the conformation in the cryo-ET reconstruction of the Sec61-TRAP-OSTA-translocon, where the Sec61 complex adopts an open plug and the lateral gate accommodates a helical peptide (Gemmer et al, 2023). Thus, the fully closed Sec61 conformation is a characteristic of the MPT in the native membrane.

### PAT and TRAP are variable MPT components

To reveal potential substoichiometric components or conformational heterogeneity, subtomograms depicting ribosome-MPT particles were subjected to two independent rounds of 3D classification focused either on the PAT complex or on luminal densities of the TRAP and BOS complexes. This classification procedure yields four classes with TRAP and PAT as substoichiometric components of MPT. Quantification hence revealed four distinct MPT variants that are distinguished by the presence (+) or absence (–) of TRAP and PAT, respectively: $MPT_{TRAP+/PAT-}$ (27%), $MPT_{TRAP-/PAT+}$ (29%), $MPT_{TRAP+/PAT+}$ (15%), and $MPT_{TRAP-/PAT-}$ (29%) (Fig 1D). Of note, BOS is not detected in substoichiometric amounts of the MPT, whereas substoichiometric association of GEL would be difficult to detect because of its integration into the lipid membrane. The finding of partial dissociation of TRAP and BOS from the MPT is consistent with a recent cryo-EM single particle analysis of affinity-isolated RNC–MPT complexes reported during the revision of this manuscript (Lewis et al, 2024).

Interestingly, we observed a slight but significant reduction of PAT levels in the presence of TRAP ($P = 7.14 \times 10^{-12}$) and, vice versa, a reduction of TRAP in the presence of PAT ($P = 8.52 \times 10^{-11}$) (Fig 2A and B). We had previously knocked out CCDC47, a subunit of the PAT complex, to identify the MPT in tomograms of microsomes isolated from ΔCCDC47 cells (Gemmer et al, 2023). To further investigate the correlation of TRAP and PAT in their association with the MPT, we now revisited this dataset using subtomogram classification. Remarkably, in the ΔCCDC47 microsomes, we did not observe a TRAP-less MPT population (Fig 2C and D). Taken together, TRAP is visible as a stoichiometric component of the MPT in the absence of PAT.

### PAT abundance decreases and TRAP occupancy increases in non-translating ribosome–MPTs

To explore the translation activity-dependent recruitment of TRAP and PAT, we analyzed the distribution of MPT classes in the context of the translation state of their bound ribosomes. We previously separated translating and non-translating ribosomes using extensive classification of ribosomal intermediate states (Gemmer et al, 2023). Translating ribosomes could be assigned to the elongation cycle, whereas most non-translating ribosomes displayed the factor CCDC124 that is associated with hibernation (Wells et al, 2020). Strikingly, in MPT particles associated with non-translating ribosomes, we observed a strong increase of the $MPT_{TRAP+/PAT-}$ from 23% to 47%, accompanied by a strong reduction of $MPT_{TRAP-/PAT+}$ from 31% to 13% compared with MPTs associated with translating ribosomes (Fig 2C and D). Hence, the absence of PAT from the MPT appears to correlate with translational inactivity of the bound ribosome, whereas TRAP abundance increases for non-translating ribosome–MPTs.

Upon knockout of the PAT subunit CCDC47, we do not observe a significant effect on ribosomal translation activity (Fig 2C). Thus, the observed stoichiometric presence of TRAP in the ribosome-MPT in the ΔCCDC47 sample is not an indirect effect of reduced translational activity.

In cells, the amount of non-translating ribosomes bound to the ER dramatically increases as a result of the unfolded protein response (UPR) (Stephens et al, 2005; Walter & Ron, 2011). Indeed, we could previously observe the dramatic accumulation of non-translating ribosomes bound to translocons in cryo-ET analysis of vesicles isolated from HEK cells that were treated with the ER stress-inducing drug dithiothreitol (DTT) at 10 mM concentration for 2 h (Gemmer et al, 2023). To clarify the composition of MPTs bound to UPR-induced translationally inactive ribosomes, we classified the subtomograms focused on TRAP and PAT (Fig 2C). Remarkably, we found TRAP to be stoichiometric in the MPT associated with the almost exclusively non-translating ribosome (97%) (Gemmer et al, 2023), whereas PAT was not detected at all (Fig 2C and D). Although UPR and DTT treatment have effects on the cell beyond reduction of translation, including up-regulation of expression of all four TRAP subunits (Nagasawa et al, 2007), which may also contribute to remodeling of ribosome-bound MPTs, the stoichiometric presence of TRAP and the absence of PAT is consistent with the trend observed in samples from untreated cells. Studies with different translation inhibitors and duration will be required to study the kinetics of dissociation in more detail. Collectively, our result suggests that PAT preferentially binds to MPTs associated with translating ribosomes and that TRAP can dissociate from PAT-containing MPTs.

The subtomogram analysis indicates that TRAP does not bind to a fraction of ribosome–MPT particles in a structurally well-defined manner, which does not necessarily mean that TRAP completely dissociates from ribosomes. Recently, a cryo-EM single particle analysis study revealed the architecture of the solubilized, isolated, ribosome-bound Sec61-TRAP translocon (Jaskolowski et al, 2023). The high resolution of the ribosome-TRAP interface in this map allowed identification of a short cytosolic, C-terminal stretch of the TRAPα subunit anchored to the 5.8S ribosomal RNA (rRNA) of the large ribosomal subunit. Consistently, we spot this anchor in subtomogram averages of membrane-bound ribosome classes (Sec61-TRAP-OSTA), but not in a soluble population revealed in our previous analysis (Gemmer et al, 2023) (Fig 3A). Interestingly, we also observe a small density colocalizing with the TRAPα anchor in multipass variants regardless of the association of the core TRAP

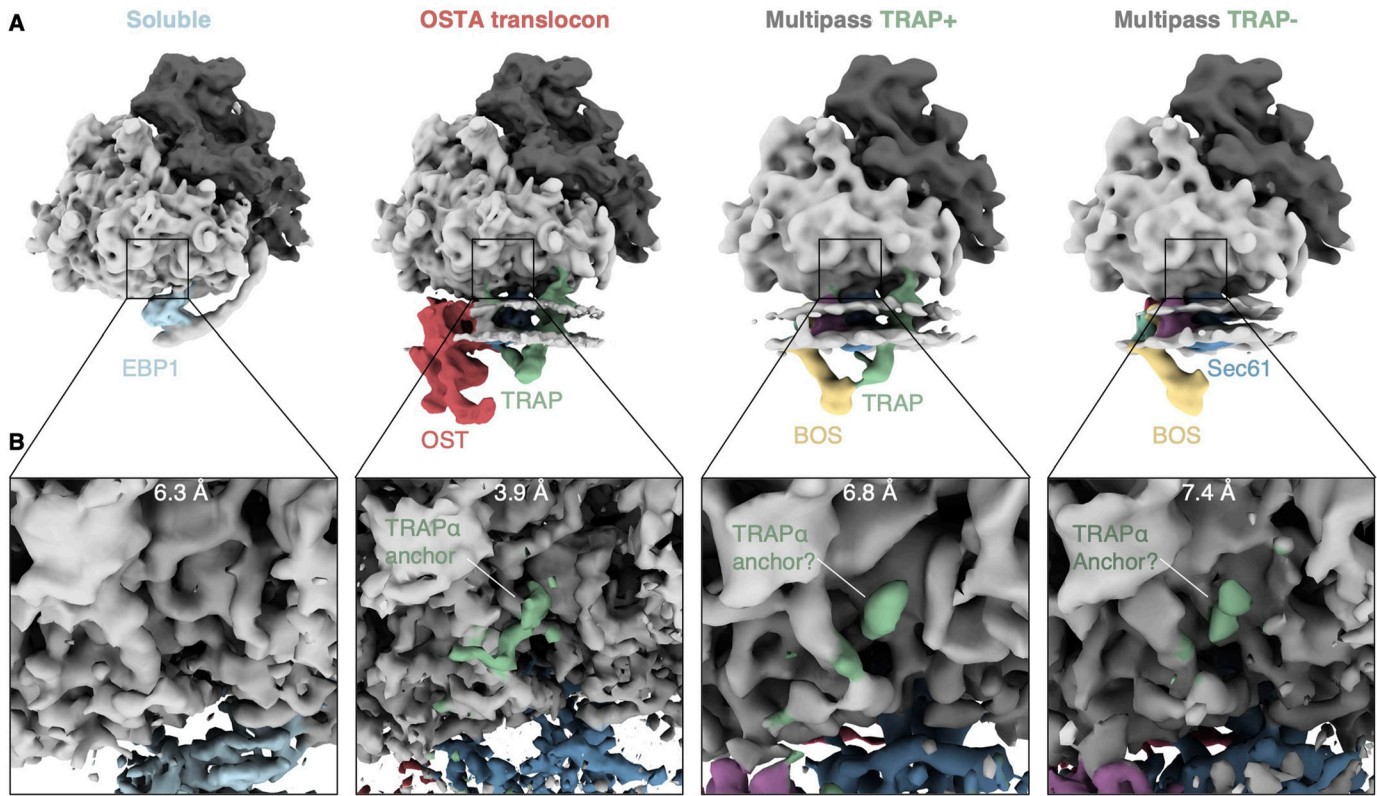

**Figure 3. Association of the TRAPα anchor domain with different ribosome populations.**
**(A)** Back side of the ribosome associated with different soluble or ER translocon variants. Reconstructions of soluble and oligosaccharyltransferase A translocon-bound ribosomes are filtered to 12 Å and multipass-bound ribosomes to 20 Å. **(B)** Close-up views of the TRAPα anchor binding site. Reconstructions are filtered to their overall resolution as indicated.

complex (Fig 3B). Although the resolution is not sufficient to unambiguously identify this density, it co-localizes with the TRAPα anchor assigned in the higher resolution maps of the isolated, ribosome-bound Sec61-TRAP translocon (Jaskolowski et al, 2023). Consistent with our observation, this previous study proposed that the TRAPα anchor flexibly tethers the complex to the ribosome even when the transmembrane and luminal TRAP segments detach from the ribosome–translocon complex (Jaskolowski et al, 2023).

The finding of partial dissociation of TRAP from MPTs associated with translating ribosomes is consistent with recent in-depth cryo-EM single particle classification of detergent solubilized substrate-engaged ribosome-MPT published in a preprint during the revision of this manuscript (Lewis et al, 2024). In this study, the occupancy of the complete TRAP heterotetramer is also low and many particles display only TRAPα binding. It is possible that the MPT$_{TRAP-}$ class in our study also contains a population with TRAPα because this single subunit is relatively small for classification of subtomograms. While TRAP dissociation from the MPT might be initiated by competition with BOS (Lewis et al, 2024), the stoichiometric binding of TRAP in ΔCCDC47 cells clearly shows that PAT is required for complete dissociation (Fig 2D).

Our results indicate a negative correlation of TRAP and PAT in the MPT, which depends on the translational activity of the ribosome. Given that PAT correlates with actively translocating ribosomes, it is plausible that its recruitment is dependent on the presence of nascent multispanning membrane proteins. Biochemical analysis of MPT assemblies engaged with defined insertion intermediates comprising TMHs with exposed hydrophilic residues suggest that the complete MPT, including TRAP and PAT, is assembled in response to a TMH substrate (Smalinskaité et al, 2022; Sundaram et al, 2022). The PAT complex subunit Asterix, an intramembrane chaperone, then specifically engages TMHs with exposed hydrophilic residues and releases its substrates upon correct folding (Chitwood & Hegde, 2020). Most PAT-less MPTs are thus likely not actively involved in substrate processing and may result from partial dissociation of PAT at later stages of the translation process, and after termination of translation. Although we observe MPTs associated with non-translating ribosomes in the native membrane, MPTs were not observed bound to non-translating ribosomes after solubilization and fractionation (Smalinskaité et al, 2022), which we attribute to the preparation causing disassociation unless the complex is stabilized by an engaged substrate. This situation is analogous to TRAP-Sec61-OSTA translocons, which typically also require a substrate for successful co-purification with ribosomes (Braunger et al, 2018) while they frequently associate with non-translating ribosomes in the native membrane (Pfeffer et al, 2015; Gemmer et al, 2023).

On the opposite side of the Sec61 complex, we find partial dissociation of TRAP from PAT-containing MPTs, which contrasts with other translocon variants (Sec61-TRAP, Sec61-TRAP-OSTA),

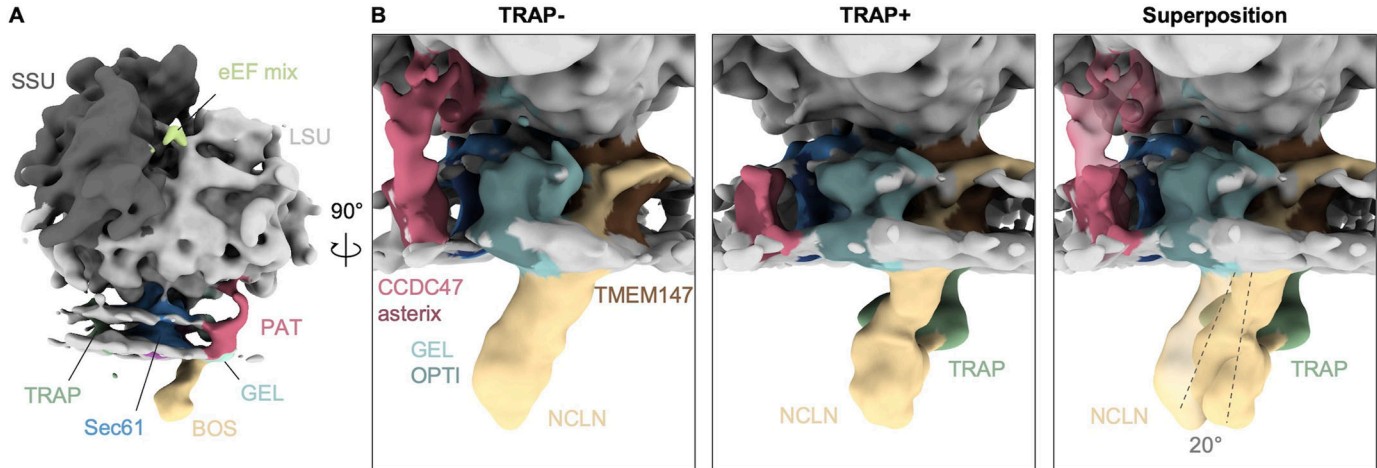

**Figure 4. Orientation of the back-of-Sec61 complex in different populations.**
**(A)** Overall structure of the ribosome-bound multipass translocon filtered to 20 Å. **(B)** Close-up side views of the translocon-associated protein (TRAP)-lacking (TRAP–) and TRAP-containing (TRAP+) multipass translocon, and their superposition with TRAP in transparent. The longitudinal axis of back-of-Sec61 is indicated as dashed line.

where TRAP is strictly stoichiometric (Pfeffer et al, 2015). A potentially functionally important consequence of TRAP binding is the modulation of the surrounding lipid membrane (Karki et al, 2023). TRAP is essential for processing of many substrates because it facilitates signal peptide insertion (Fons et al, 2003; Nguyen et al, 2018), interacts with translocated substrates (Gorlich et al, 1992), and contributes to safeguarding membrane protein topogenesis (Sommer et al, 2013). Thus, tight TRAP association is likely important in early stages of substrate processing, including signal peptide insertion. In the context of the MPT, one possible scenario is that disassociation prevents TRAP-assisted insertion of TMHs into the Sec61 lateral gate at later stages of TMH processing.

## TRAP interacts with the BOS complex

The molecular mechanism underlying TRAP disassociation from the MPT remains unclear. It has been suggested that the luminal segment of the BOS complex may displace TRAP at the pore exit of Sec61 (Smalinskaitė et al, 2022). To examine the structural organization of the two accessory factors in the native membrane, we obtained merged TRAP containing MPTs (MPT$_{TRAP+/PAT-}$ and MPT$_{TRAP+/PAT+}$, 8,477 particles) and TRAP-deficient MPTs (MPT$_{TRAP-/PAT-}$ and MPT$_{TRAP-/PAT+}$, 6,194 particles). In the absence of TRAP, the luminal segment of BOS adopts a tilted conformation with respect to the membrane (Fig 4A). This conformation is consistent with the orientation observed in the cryo-EM structure of the TMCO1-affinity purified ribosome-MPT complex (EMD-21435 [McGilvray et al, 2020]), which displays the TRAP complex in an unique position distant from BOS and Sec61. In the presence of TRAP, BOS undergoes a ~20° rotation and projects almost orthogonally from the membrane, in line with a cryo-EM structure of the MPT, in which TRAP remained bound to the MPT (EMD-26133 [Smalinskaitė et al, 2022]) (Figs 4B and S3A and B). Intriguingly, BOS would clash with the luminal domain of TRAPα when adopting the tilted conformation, as noted also in Lewis et al (2024). Thus, conformational switching of BOS could trigger the release of TRAP from the MPT, as suggested earlier

(Smalinskaitė et al, 2022; Lewis et al, 2024). However, the steric clash does not explain the role of the PAT complex in TRAP disassociation, which is most evident from the fact that we observe TRAP stoichiometrically in ribosome–MPT complexes from ΔCCDC47 cells. TRAP and PAT reside at opposite sides of Sec61 and do not display any notable clashes (Fig 1), precluding an obvious structural explanation for the negative correlation of PAT and TRAP in the MPT.

In more detail, TRAP and BOS compete for the hinge loop of Sec61 (Lewis et al, 2024). We previously suggested that the interaction of the luminal TRAPα domain with the hinge loop plays an important role for Sec61 lateral gate opening (Gemmer et al, 2023), and thus might contribute to Sec61 being in an closed conformation in MPT, while it adopts an open lateral gate in all other translocons in the native ER membrane. Another recently emerging player in the regulation of lateral gate opening and TRAP dissociation is RAMP4/SERP1, which may bind to the lateral gate of TRAP-Sec61 and TRAP-Sec61-OSTA translocons, as suggested by AlphaFold models (Gemmer, 2023; Lewis et al, 2024). Further experiments will be required to determine the molecular mechanism of TRAP release and Sec61 lateral gate switching in the context of MPT.

Subtomogram recentering and local refinement focused on the luminal densities of TRAP and BOS yielded notably improved densities with a resolution of ~15 Å (Fig 5A). Whereas the atomic models of NCLN (AF-Q969V3) and TRAP (PDB-8B6L) could be fitted into the recentered reconstruction, two small densities, each with an approximate molecular weight of 10 kD, remain unexplained. Both densities are bound to the luminal domain of NCLN, whereas the membrane-proximal density interacts with the luminal domains of TRAPβ and TRAPδ (Fig 5A).

An obvious candidate for the unidentified densities is NOMO, a subunit of the BOS complex (Haffner et al., 2004, 2007; Dettmer et al, 2010) because it was shown to be highly abundant in isolates (McGilvray et al, 2020). Large stretches of the three NOMO paralogs in the human genome are identical, which makes it challenging to specify the relative expression levels of specific paralogs by mass spectrometry. NOMO is a 134-kD ER-resident single-pass membrane

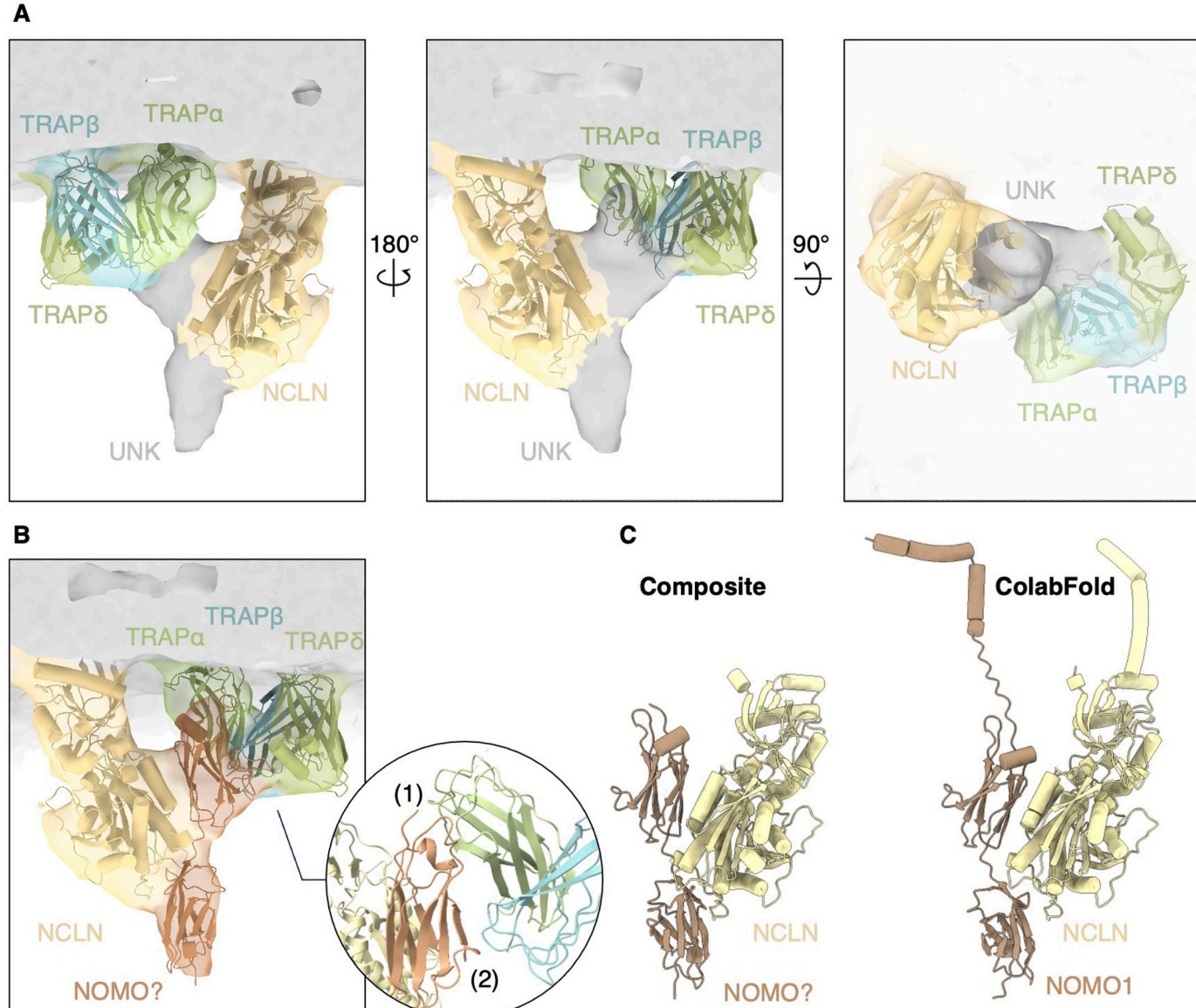

**Figure 5. Translocon-associated protein (TRAP) interacts with the NCLN–nodal modulator (NOMO)–TMEM147 complex.**
**(A)** Recentered and locally refined reconstruction of the TRAP-containing multipass translocon. Luminal domains of TRAP (8B6L, green) and NCLN (AF-Q969V3, yellow) fit well into the density map. Two unidentified densities (UNK, grey) have a molecular weight of 10 kD each. **(B)** Molecular model of the C-terminal Ig-like domain 11–12 of NOMO1 (AF-Q15155) placed into the density map of the unidentified densities. The inset shows a close-up view of the putative interaction site between NOMO and TRAP. NOMO residues 1,068–1,072 and 1,117–1,120 likely interact with TRAPα residues 90–94 and 191–195 (1), and NOMO 1,084–1,089 with TRAPβ 51–53 and 83–86 (2). **(B, C)** Comparison of our model based on the reconstruction from (B) and the ColabFold prediction model of the NCLN–NOMO1 complex. NOMO1 domains 1–9 were removed for clarity.

protein with a 12- kD luminal domain. NOMO and NCLN were shown to associate independently of TMEM147 and the domains that mediate their interaction reside in the ER lumen (Dettmer et al, 2010). Despite its abundance in the isolate, NOMO was not modelled into the single particle analysis reconstruction, presumably because the resolution of ER luminal densities was affected by flexibility (McGilvray et al, 2020). Negative stain electron microscopy and sequence analysis of NOMO suggest an elongated shape, comprising 12 immunoglobulin (Ig)-like domains that are arranged like "beads on a string" (Amaya et al, 2021). AlphaFold models of

NOMO paralogs are consistent with this notion (Jumper et al, 2021; Varadi et al, 2022). The two most C-terminal NOMO domains, which reside adjacent to the ER membrane-anchoring TMH, are prime candidates to explain the unknown densities and indeed fit well into the low-resolution cryo-ET reconstruction without major clashes to NCLN or TRAP (Fig 5B). They contact TRAP at the hydrophobic cradle formed by the luminal domains of TRAPα, TRAPβ, and TRAPδ, which was hypothesized to interact with translocating substrates (Jaskolowski et al, 2023). Thus, NOMO may occupy this site preferably at MPT complexes associated with non-translating

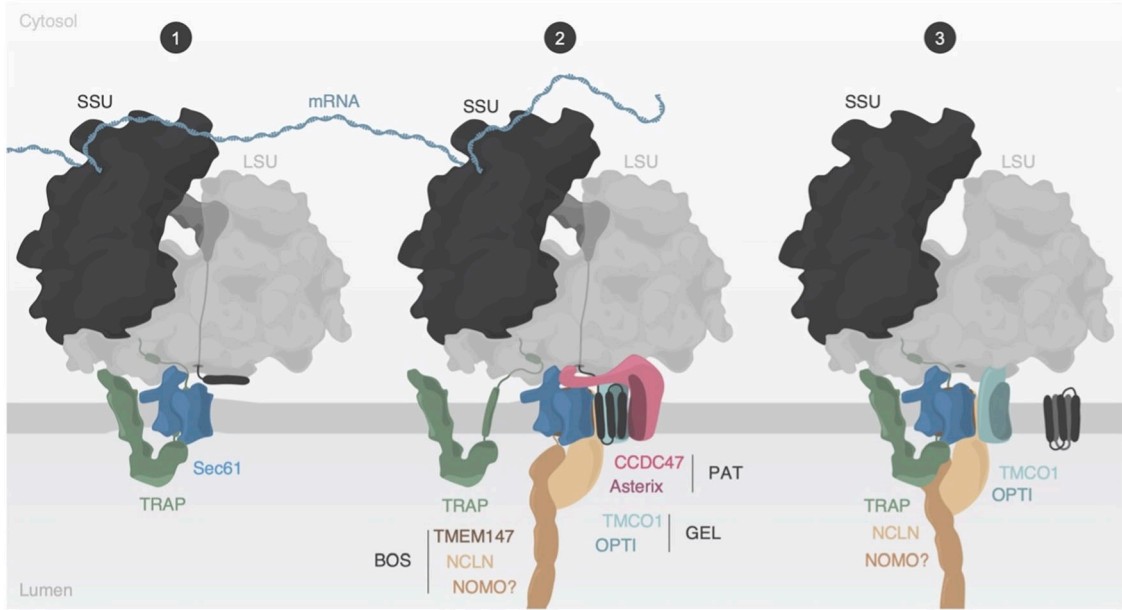

**Figure 6. Model of multipass membrane protein biogenesis.**
(1) Synthesis of multispanning membrane proteins is facilitated by the ER translocon-bound ribosome. The translocon-associated protein (TRAP) complex engages with the translocon during the early stage. (2) Transmembrane helices of the nascent substrate are inserted into the cavity behind Sec61, whereas Sec61's lateral gate and pore remain closed. The PAT complex engages with exposed hydrophilic areas of substrate proteins and the TRAP complex partially dissociates from the translocon likely to avoid TRAP-assisted insertion into Sec61's lateral gate. The TRAPα anchor domain may remain bound to the ribosome to retain TRAP in proximity to the translocon. (3) After substrate folding and release, PAT dissociates from the ribosome and translocon, whereas TRAP is recruited again and interacts with the back-of-Sec61 complex in the ER lumen.

ribosomes. The remaining NOMO Ig-like domains cannot be explained in the subtomogram average, which is likely due to their flexibility.

To test the hypothesis that NOMOs C-terminal Ig domains constitute the unknown density, we built a model of the NCLN–NOMO complex using Colabfold (Mirdita et al, 2022). Intriguingly, this model positions the C-terminal NOMO1 domains almost identical to our fit into the cryo-ET structure with high confidence (PAE < 10, pLDDT > 85%) (Figs 5C and S4A–E). In the model, the NCLN residues 420–458 and 478–486 interact with the NOMO1 residues 969–978, 1,027–1,033, 1,049-59, 1,097–1,099, and 1,137–1,140 located in the last two C-terminal domains. Although experimental validation of our assembly model remains, our results strongly suggests that the luminal domain of NOMO mediates association of TRAP and NCLN in the ER lumen. These conclusions are consistent with the recent preprint by Lewis et al (2024).

A direct interaction between TRAP and NOMO has not been reported to date. Interestingly, NOMO has recently also been shown to mediate interaction of the BOS complex with the EMC in a similar fashion as with TRAP (Page et al, 2023 Preprint), suggesting a more general tethering role of NOMO. In humans and *C. elegans*, NCLN, NOMO, and TMEM147 (or their homologs) have been shown to regulate levels, subcellular localization, and subunit composition of different homo- and heterooligomeric multispanning membrane proteins (Almedom et al, 2009; Rosemond et al, 2011; Kamat et al, 2014; Christodoulou et al, 2020), pointing to a role in membrane protein biogenesis.

Moreover, NOMO has recently been shown to affect ER sheet morphology. The precise functions of NOMO, NCLN, and TMEM147 and their interplay with TRAP or other translocon accessory factors, however, remain poorly understood.

### Hypothetical model for the re-arrangements of the multipass translocon

Taken together, the in-depth subtomogram classification allows devising a hypothetical model for the re-arrangements of the multipass translocon before, during, and past multipass transmembrane helix protein biogenesis (Fig 6). Our previous analysis of translocon assembly states along ER-bound polysomes suggested that ribosomes may preferentially encounter Sec61-TRAP translocons early on during biogenesis (Gemmer et al, 2023). The strong correlation of PAT with actively translating ribosomes revealed by our analysis suggests that GEL, PAT, and BOS are recruited rapidly, possibly en bloc, for the insertion of nascent multipass-transmembrane proteins as translation proceeds. TRAP is detected less frequently in the MPT during translation, which is likely not due to complete disassociation, but rather due to increased structural variability. The direct ribosome-binding modules of TRAPα and TRAPγ may keep the TRAP subunits in the proximity of the MPT. We also show that the MPT can associate with non-translating ribosomes, which likely remained bound after terminating the synthesis of multipass-transmembrane proteins. Whereas PAT tends to dissociate from the MPT when translation is not ongoing, TRAP preferentially binds

again in a structurally well-defined manner. A characteristic of the TRAP-containing MPT is a density between TRAP and BOS, which may be explained by the hitherto unresolved BOS subunit NOMO.

Remaining questions are the precise molecular mechanisms that cause the rearrangements of the ribosome-bound translocon complexes such as the negative correlation of PAT and TRAP in the MPT. Furthermore, the role of different translocon complexes bound to non-translating ribosomes and their role for initial stages of translation, also upon the unfolded protein response, is a largely uncharted territory that may be studied best in the native membrane context in the future.

# Materials and Methods

## Sample preparation and cryo-ET data processing

Cell culture, sample, and grid preparation, data collection and processing, tomogram reconstruction, and particle localization were described previously (Gemmer et al, 2023). In brief, HEK 293-F (R79007; Thermo Fisher Scientific) or ΔCCDC47 cells were grown in suspension in FreeStyle medium, harvested, and washed (50 ml of 0.5–1 × 10^6 cells/ml). HEK 293-F cells used for ER stress studies were treated with 10 mM DTT for 2 h before harvesting. Cells were resuspended in lysis buffer (2–4 ml, 10 mM HEPES-NaOH, pH 7.4, 250 mM sucrose, 2 mM MgCl₂, 0.5 mM DTT, and protease inhibitor cocktail [Roche]) and lysed using an Isobiotec cell cracker on ice. The lysate was cleared using a cooled tabletop centrifuge (1,500g, 2–3 × 5 min, 4°C, in 2 ml tubes). ER vesicles were pelleted (10,000g, 10 min, 4°C) and washed with resuspension buffer (10 mM Hepes, 250 mM sucrose, 1 mM MgCl₂, and 0.5 mM DTT). The pellet was resuspended at a concentration of ~50 mg/ml, frozen in liquid nitrogen, and stored at –80°C until further use. The ER vesicles were diluted in resuspension buffer to a concentration of 2–3 mg/ml, and 2 μl were applied onto a glow-discharged lacey carbon grid (Quantifoil) together with 4 μl of diluted gold fiducials (UMC Utrecht). Grids were immediately blotted from the backside for 5–6 s and plunged into a mix of liquid ethane and propane using a manual plunger.

In total 1,141 tilt series (non-stressed: 869, DTT-treated: 212, ΔCCDC47: 60) were acquired on a Talos Arctica (Thermo Fisher Scientific) operated at an acceleration voltage of 200 kV and equipped with a K2 summit direct electron detector and energy filter (Gatan). Images were recorded in movies of 7–8 frames at a target defocus of 3 μm and a pixel size of 1.72 Å on the specimen level. Tilt series were acquired in SerialEM (3.8) (Mastronarde, 2005) using a grouped dose-symmetric tilt scheme (Hagen et al, 2017) covering a range of ±54° with an angular increment of 3°. The cumulative dose of a series did not exceed 80 e⁻/Å².

## Subtomogram analysis

Subtomogram analysis of all ribosomes was described previously (Gemmer et al, 2023). In brief, the localized particles were extracted and aligned in RELION (3.1.1) (Scheres, 2012). The

aligned particles were further refined in 2–3 rounds in M (1.0.9) (Tegunov et al, 2021) focused on the LSU. Afterwards, newly extracted subtomograms were subjected to 3D classification in RELION to discard remaining false positives, poorly aligned particles, and lone LSUs. The remaining 134,350 particles were used for subsequent focused classification steps.

## Classification of ER translocon variants

134,350 ribosome particles from the non-treated ER vesicles were subjected to 3D classification (without reference, with soft mask, T = 4, classes = 20) in RELION focused on the ribosomal tunnel exit. Particles were sorted into Sec61-TRAP-bound, Sec61-TRAP-OST-bound, Sec61-multipass-bound, EBP1-bound ribosomes (soluble), and a combined class of ribosomes with ambiguous densities. Ribosomes with ambiguous densities were subjected to two further classification rounds and sorted the respective class from above until no further separation could be achieved.

We re-extracted subtomograms of the multipass translocon (14,671 particles) at a voxel size of 6.9 Å recentered onto the center of Sec61. Subsequently, subtomograms were subjected to classification focused on the ER luminal area (with reference of the entire multipass translocon population, with soft mask, T = 3, classes = 2) or focused on the cytosolic domain of the PAT complex and ribosomal tunnel exit (with reference, with mask, T = 3, classes = 2). The TRAP-containing multipass translocon was refined using local angular searches in RELION. Subtomograms were centered onto the ribosome again, re-extracted, and averaged.

## Model fitting and prediction

TRAPα, TRAPβ, TRAPδ (8B6L), NCLN (AF-Q969V3), and luminal Ig-like domains 11–12 of NOMO1 (AF-Q15155) were fitted into the density map of the recentered reconstruction of the MPT. The AlphaFold Colab (Evans et al, 2021 Preprint) model prediction was built based on sequences of human NCLN (Q969V3) and NOMO1 (Q15155). The sequence of the signal peptide was removed from NCLN and NOMO before prediction.

## Statistical analysis

The statistical analysis was described previously (Gemmer et al, 2023). Statistical significance for the fitted logistic parameters was determined with a two-sided Wald-test in R 3.6.1 and mclogit. P-values were adjusted for multiple comparisons with the Hochberg method.

# Data Availability

We made use of previously published atomic models from the PDB (8B6L, 6W6L, 7TUT) and the AlphaFold Protein Structure Database (Q969V3, Q15155). Raw data of ER microsome tilt series are deposited on EMPIAR-11751.

# Supplementary Information

# Acknowledgements

This work was supported by the European Research Council under the European Union's Horizon 2020 Program (ERC Consolidator Grant Agreement 724425—BENDER) and the Nederlandse Organisatie voor Wetenschappelijke Onderzoek (Vici 724.016.001, research program TA 741.018.201, and National Roadmap for Large-Scale Research Infrastructure NEMI 184.034.014).

## Author Contributions

M Gemmer: conceptualization, formal analysis, investigation, visualization, and writing—original draft, review, and editing.
ML Chaillet: formal analysis and investigation.
F Förster: conceptualization, resources, supervision, investigation, project administration, and writing—original draft, review, and editing.

## Conflict of Interest Statement

The authors declare that they have no conflict of interest.

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
