## [Reviewer comments · Life Science Alliance]

Life Science Alliance

Exploring the molecular composition of the multipass translocon in its native membrane environment

Maximilian Gemmer, Marten Chaillet, and Friedrich Förster

DOI: <https://doi.org/10.26508/lsa.202302496>

Corresponding author(s): Friedrich Förster, Utrecht University and Maximilian Gemmer, Utrecht University

Review Timeline:

Submission Date:	2023-11-27
Editorial Decision:	2024-01-08
Revision Received:	2024-05-09
Editorial Decision:	2024-05-15
Revision Received:	2024-05-22
Accepted:	2024-05-23

Transaction Report:

January 8, 2024

Re: Life Science Alliance manuscript #LSA-2023-02496-T

Friedrich Foerster
Bijvoet Center for Biomolecular Research
In Situ Structural Biology
Padualaan 8
Utrecht 3584 CH
Netherlands

Dear Dr. Foerster,

Thank you for submitting your manuscript entitled "Exploring the molecular composition of the multipass translocon in its native membrane environment" to Life Science Alliance. The manuscript was assessed by expert reviewers, whose comments are appended to this letter. We invite you to submit a revised manuscript addressing the Reviewer comments.

Thank you for this interesting contribution to Life Science Alliance. We are looking forward to receiving your revised manuscript.

Sincerely,

B. MANUSCRIPT ORGANIZATION AND FORMATTING:

Reviewer #1 (Comments to the Authors (Required)):

This short Ms by Gemmer et al. is a follow-up on their recent publication in Nature (2023), where they used cryo-ET and subtomogram averaging on ER-derived rough microsomes to visualize various ribosome-Sec61 complexes involved in protein secretion and membrane protein biogenesis. In the present Ms, they perform a deeper analysis of what appears to be the same data set, focusing on the "multipass translocon" complex that is involved in the biosynthesis of multipass membrane proteins.

The main findings are:

- Sec61 is in the closed state in the multipass translocon complex (Sec61-PAT-TRAP-BOS), but not in the Sec61-TRAP-OSTA complex.
- In multipass translocon complexes, there is a slight inverse correlation between the presence of PAT and TRAP.
- CCDC47 is part of PAT. When CCDC47 is knocked out, only TRAP is found bound to Sec61. Similarly, with DTT treatment, only TRAP is found bound to Sec61.
- TRAP is found more on inactive ribosomes and PAT is found more on active ones.
- BOS is tilted when TRAP is not there, the tilt would cause it to clash with TRAP α .
- NOMO probably accounts for the extra density that bridges between NCLN and TRAP.

Comments:

- It appears that no new data were collected for this study, but the text is not totally clear on this point. It should be unequivocally stated whether or not new samples were prepared and new cryo-ET data were collected for the present study, or whether it is based only on further processing of the already published data set.
- The image processing is described only in outline. A Cryo-ET workflow figure might help the reader understand how the population distribution is obtained and at which point are the different particle centering operations done, when centering at the ER region is used and when re-centering around the ribosome is done. Maybe also indicate the masks used for focused classification.
- Is the resolution sufficient to unambiguously tell whether Sec61 is open or closed?
- It seems that BOS is present in all the different multipass translocon classes. Perhaps this should be pointed out?
- Tilting of BOS might be influenced by membrane thinning, which has been shown to be associated to the presence of TRAP by Karki et al. (EMBO Rep Dec 2023). Might be worth citing this paper.
- Supplementary Fig. S3 does not seem to be mentioned in the text. This should be done, or the Figure should be removed. It is not clear which particles were used for the different reconstructions in this Figure. The two on the right correspond to WT and the CCDC47 knockout? Or are they all classes from "WT"?
- The very same day that this Ms was deposited in BioRxiv (Nov 29), a Ms from the Voorhees lab was also deposited in BioRxiv (<https://www.biorxiv.org/content/10.1101/2023.11.28.569054v1>). The findings in this latter paper are of clear relevance to the present study, and should be discussed.
- The Conclusions section is vague and a bit disappointing, and could perhaps be made more interesting by being a bit more explicit in discussing possible mechanistic interpretations of the data.
- The Ms adds some new information on the various subcomplexes in the multipass translocon and would be suitable as a short communication.

Minor issues:

- Figs 2a and 2b would benefit from a legend.
- In the "Data availability" section: "We made use of a previously published atomic models..." (remove "a")
- In the first paragraph of the results subheading "TRAP interacts with the BOS complex", in the second-to-last sentence, the first words "Vice versa, however" are awkward. The readability might benefit by being replacing that phrase with something simpler, like "alternatively."
- In the third paragraph of the results subheading "TRAP interacts with the BOS complex": "An obvious candidate for the unidentified densities is NOMO, a subunit of the BOS complex (citation), because it was shown..." (add "it")

Reviewer #2 (Comments to the Authors (Required)):

This study describes additional analysis of cryo-ET data recently published in PMID 36697828. The authors focus on the TRAP complex, implicated in facilitating translocation through the Sec61 channel, and its relationship to the multipass translocon (MPT) subcomplexes known as PAT, GEL, and BOS. The main conclusions drawn by the authors are that TRAP and BOS contact each other, potentially through the NOMO subunit of BOS, and that this interaction may influence the relative positions of the TRAP and BOS complexes.

The advance here is rather modest, and it is difficult to draw any appreciable functional implications of the observations or propose any specific hypotheses related to either TRAP or BOS function. This is evident in the "Conclusions" section, which is not able to articulate what was actually advanced by this study beyond rather superficial and vague statements. The main useful contribution beyond the TRAP-BOS relationship is adding to the census of translocon types in native membranes based on classification of the particles observed by cryo-ET. If one were to publish this analysis, the following suggestions for improvement (in the order they appear in the paper) should probably be incorporated or addressed.

1. The authors are vague about how they are defining the MPT. For example, in the abstract they talk about PAT associating with the MPT, but as per earlier definitions, PAT is an integral component of the MPT. Perhaps for simplicity, the authors are calling all complexes containing BOS and lacking OSTA the MPT, as this can be readily distinguished in the cryo-ET data? Regardless, the authors need to define their terms and use them consistently.
2. The authors talk about active and inactive translocons. What does 'active' mean, and how are they defining this? I assume they cannot see substrate, so this cannot be the deciding parameter. Also, it presumably does not reflect the open or closed state of Sec61. Perhaps they are talking about whether the associated ribosome is translating or not? If so, then they should be precise and say something like "...translocons associated with translating ribosomes showed... whereas translocons associated with non-translating ribosomes showed..." I realise it is cumbersome, but the lack of precision in how they describe the findings makes the paper very difficult to understand, even by an expert. It is also important to be clear about exactly what metric is used to assign something as translating or not, active or inactive, etc.
3. Still on terminology, how is hibernating defined, and why use this rather than inactive? I think hibernation means something different than inactive, and it is not clear why hibernating ribosomes would associate with translocons anyway.
4. In the abstract, the conclusion that TRAP associates with inactive translocons is going to be very misleading. There is little doubt that TRAP is part of translocating Sec61 complexes (plenty of past biochemical and structural work to support this), and indeed the authors know this as they talk about the TRAP cradle associating with substrates and so forth.
5. The persistent use of native or near-native is again not precise or helpful. The fact that microsomes are isolated necessarily means an important aspect of the sample is not native or near-native - there are countless things that can happen during this biochemical process including dissociation of key factors, loss of tRNA due to hydrolysis, and so forth. I therefore object to such terms. What is native is the membrane environment, and the authors should strive to be precise about this. Again, I release it is cumbersome terminology, but it will also minimise confusion in the future.
6. The authors should note the caveat that the ribosome-nascent chain state might change during fractionation, altering to some degree the correlation between what is seen at the ribosome (i.e., translating or not) and what is seen in the membrane. This is important, because some of the claims are at odds with earlier conclusions. For example, work by Smalinskaite et al. and Sundaram et al. indicates that the MPT is assembled in response to substrate, and the data supporting this seems fairly clear. So one would not expect to see the MPT associated with non-translating ribosomes, so observing this population warrants some explanation. Being explicit about the caveats will be helpful in providing some possible resolutions to this difference in observations.
7. Introduction, paragraph 1: the sentence starting "Sec61 assembles with..." is redundant with the previous sentence and can be deleted.
8. Introduction, paragraph 2: the GET pathway references are not appropriately chosen. The initial discovery that post-translational targeting of TA proteins is mediated by GET3 was reported by Stefanovic & Hegde, 2007, Cell. The demonstration that the membrane subunits directly mediate TA protein insertion was first shown in reconstitution experiments by Mariappan et al., 2011, Nature and Wang et al., 2011, Mol. Cell. The Schuldiner paper could not really draw conclusions about insertion, only that GET1/2 are a receptor for GET3. The role of GET as an insertase was shown by Wang et al., 2014, Nature.
9. Introduction, paragraph 2, last sentence: affinity tagged TMCO1 was only part of the strategy; the strategy used by Smalinskaite et al. and Sundaram et al. was affinity tagged defined membrane protein insertion substrates. Probably best to split into two sentences and clarify what was done in which study.
10. Introduction, paragraph 3, line 2: "complexes" should be "complex".
11. Introduction, paragraph 3: The GEL complex was not defined by McGilvray et al., 2020, as they did not know about OPTI. The other references cited are appropriate (Lewis, Smalinskaite, Sundaram).
12. Introduction, paragraph 3: The finding that CCDC47 impedes Sec61 opening was shown in Smalinskaite et al., 2022, not in either Chitwood or Meacock.
13. Introduction, paragraph 3: Meacock cannot realistically be cited for the PAT complex. They of course did pioneering work in detecting a crosslink near membrane protein insertion intermediates, but did not know what that crosslink was, what its function was, or that it had CCDC47 as a partner.
14. Introduction, paragraph 3: McGilvray didn't have any functional data about what the multipass translocon does. The role in

TMH insertion was shown later in the 2022 Nature papers.

15. Introduction, paragraph 3, last sentence: this is potentially misleading. Sundaram certainly examined the activity-dependent compositional variation, and to a lesser extent, so did Smalinskaite. Both these studies clearly established substrate-triggered MPT assembly and disassembly. I think the authors mean to emphasise the native membrane part of the sentence, so it should be re-phrased to focus on this.

16. Pg. 5, paragraph 1: the conclusion that TRAP is recruited to inactive translocons is incompatible with the findings that TRAP in fact is needed during translocation (e.g., to initiate the process by facilitating signal sequence insertion; Fons et al., 2003, JCB), and also crosslinks to translocating polypeptides (e.g., Gorlich et al., 1992, Nature). There is also earlier structural data on the TRAP complex and this too suggests it associates with translating and translocating translocons. The authors need to both be careful about terminology and also discuss the relationship of this work to earlier studies.

17. Using DTT as a surrogate for modulating translation is just an awful idea as DTT will have many, many other effects. I think this needs to be acknowledged. Why they don't use a translation initiation inhibitor is unclear, but regardless, they should acknowledge the pleiotropic effects of DTT. Also, mention the time and concentration of treatment in the Results so one does not have to hunt down this key piece of information.

18. The authors might wish to discuss and incorporate into their thinking the new preprint by Lewis and Hegde (BioRxiv). This shows that TRAP-alpha can remain associated whereas the membrane regions of the other subunits are disordered due to the luminal domain being shifted by the BOS complex.

19. Pg. 5, bottom: The "substrate-dependent recruitment behavior of PAT during translocation" was also shown in Sundaram et al., 2022.

20. Pg. 6, top: McGilvray didn't even know about Asterix, so it is unclear why this is cited for function of the PAT complex. The appropriate citation is Chitwood & Hegde, 2020, Nature.

21. Pg. 6, top: The sentence "we speculate that PAT associates..." is not really speculation. This was shown directly in Smalinskaite et al., 2022, but mutating the hydrophilic residues of the substrate TMH and analyzing the consequences for MPT assembly.

22. The model by McGilvray (PDB: 6W6L) lacks the TRAP complex, but the authors should probably have a careful look at the maps as I believe it does contain the TRAP complex as evident by its prominent luminal domain.

23. Pg. 6 bottom: mass spectrometry probably cannot distinguish between NOMO1, 2 and 3, so perhaps re-phrase that sentence. The supposed enrichment of NOMO1 and 3 is probably just an annotation issue.

24. Pg. 7, bottom: I cannot see how the word "comprehensive" is justified in the Conclusions section. The overall number of particles is really too small to do thorough classification and it is worth being a bit less bombastic and a bit more realistic here.

Reviewer #3 (Comments to the Authors (Required)):

Review of manuscript of Gemmer et al. entitled: 'Exploring the molecular composition of the ER multipass translocon in its native membrane environment'

This manuscript describes the cryo-electron tomography structure characterization of co-translational ribosome/Sec61 complexes together with the translocon associated protein complex (TRAP) and components of the recently described multipass translocon and the intramembrane chaperone PAT. The work builds upon previous recent work from the same group (Gemmer et al, 2022) and recent body of work describing functions and structures of different Sec61 complexes involved in insertion of multipass membrane proteins. Collectively, the latter complexes are termed the multipass translocon, which comprises of three subcomplexes (GEL, PAT and BOS complexes).

The work presented here comes from further analysis and classification of the compositionally diverse multipass translocons, using data collected in the earlier Gemmer et al. study. Specifically, the authors expand the analysis of Sec61 recruitment of TRAP versus components of the multipass translocon on Sec61 and demonstrate that TRAP appears to preferentially associate with Sec61 bound to inactive ribosomes, whereas multipass translocon associated with actively translating ribosomes.

A new finding is identification of an unknown density in Sec61 complexes containing TRAP and the BOS subcomplex of the multipass translocon. The authors speculate that this density would represent membrane-proximal subunit of the NOMO subunits of the BOS complex, although the low resolution of the experimental density precludes unambiguous assignment. A model built with Colabfold fits well to the observed density, providing support to the proposed identity of NOMO in this location bound to the ER luminal domains of TRAP.

Major points:

1. The resolution of the reconstruction is quite modest, which prevents unambiguous assignment of the putative NOMO Ig-like domains. I realize that testing the Colabfold-derived model of NOMO-NCLN remains a future task, it would be important to indicate what the main proposed contacts in the Colabfold model are.

Minor points:

1. The authors suggest that TRAP would preferentially associate with inactive translocons based on the finding that TRAP is preferentially enriched with inactive ribosomes in the cryo-ET analysis. Is it believed that TRAP would associate with Sec61 already before ribosome docking?
2. Page 7. The authors state that "our results strongly suggests that the luminal domain of NOMO mediates association of TRAP and NCLN in the ER lumen." Does the current placement of the Colabfold-derived model of NOMO-NCLN suggest what the interaction site on TRAP would be? It appears that this interface would involve residues of TRAP and TRAP . Could the authors indicate the TRAP putative interaction surface and highlight some putative interacting residues of the TRAP luminal domains? This would ideally take place as a new additional figure panel.

Point-by-point reply

We thank the reviewers for carefully reading our manuscript and for their thoughtful comments. We extensively revised the manuscript addressing this feedback. Major changes include:

- We clearly distinguish between the activity of MPT and translation activity of the associated ribosome.
- The conclusion section has been re-written
- We include recently published preprints in the discussion of our results

Below we address the points raised by the reviewers and the resulting changes of the manuscript point by point (our response in blue):

Reviewer #1:

This short Ms by Gemmer et al. is a follow-up on their recent publication in Nature (2023), where they used cryo-ET and subtomogram averaging on ER-derived rough microsomes to visualize various ribosome-Sec61 complexes involved in protein secretion and membrane protein biogenesis. In the present Ms, they perform a deeper analysis of what appears to be the same data set, focusing on the "multipass translocon" complex that is involved in the biosynthesis of multipass membrane proteins.

The main findings are:

- Sec61 is in the closed state in the multipass translocon complex (Sec61-PAT-TRAP-BOS), but not in the Sec61-TRAP-OSTA complex.
- In multipass translocon complexes, there is a slight inverse correlation between the presence of PAT and TRAP.
- CCDC47 is part of PAT. When CCDC47 is knocked out, only TRAP is found bound to Sec61. Similarly, with DTT treatment, only TRAP is found bound to Sec61.
- TRAP is found more on inactive ribosomes and PAT is found more on active ones.
- BOS is tilted when TRAP is not there, the tilt would cause it to clash with TRAP α .
- NOMO probably accounts for the extra density that bridges between NCLN and TRAP.

Comments:

- It appears that no new data were collected for this study, but the text is not totally clear on this point. It should be unequivocally stated whether or not new samples were prepared and new cryo-ET data were collected for the present study, or whether it is based only on further processing of the already published data set.

We clarified that previously acquired data were re-analyzed in the last paragraph of the introduction section.

- The image processing is described only in outline. A Cryo-ET workflow figure might help the reader understand how the population distribution is obtained and at which point are the different particle centering operations done, when centering at the ER region is used and when re-centering around the ribosome is done. Maybe also indicate the masks used for focused classification.

We provide supplementary Figure 1, which outlines the processing.

- Is the resolution sufficient to unambiguously tell whether Sec61 is open or closed?

The resolution is sufficient to assign the Sec61 of the MPT because helices can be resolved. Correlation coefficients with open and closed atomic models are also clearly distinct, which we provide in the revised caption of Supp. Figure 4. Correlation coefficients of open (Sec61 $\alpha\beta\gamma$ from PDB-7TUT) 0.8541 vs closed (Sec61 $\alpha\beta\gamma$ from PDB-3JC2) 0.8815 (filtered to 10Å).

- It seems that BOS is present in all the different multipass translocon classes. Perhaps this should be pointed out?

We included the following sentence in the results section: "Of note, BOS is not detected in substoichiometric amounts of the MPT, while substoichiometric association of GEL would be difficult to detect due to its integration into the lipid membrane."

- Tilting of BOS might be influenced by membrane thinning, which has been shown to be associated to the presence of TRAP by Karki et al. (EMBO Rep Dec 2023). Might be worth citing this paper.

In the vicinity of TRAP the membrane structure is indeed altered and hence we included the reference in the functional introduction of TRAP. Nevertheless, changes of membrane structure upon BOS tilting are less obvious.

- Supplementary Fig. S3 does not seem to be mentioned in the text. This should be done, or the Figure should be removed. It is not clear which particles were used for the different reconstructions in this Figure. The two on the right correspond to WT and the CCDC47 knockout? Or are they all classes from "WT"?

We apologize for not referencing this figure in the manuscript. Figure S3 (now S2) is now referenced in the text. The names of the populations have been updated and their composition is further explained in the figure caption.

- The very same day that this Ms was deposited in BioRxiv (Nov 29), a Ms from the Voorhees lab was also deposited in BioRxiv (<https://www.biorxiv.org/content/10.1101/2023.11.28.569054v1>). The findings in this latter paper are of clear relevance to the present study, and should be discussed.

We included this work in the discussion of the role of NOMO at the end of the results and discussion section.

- The Conclusions section is vague and a bit disappointing, and could perhaps be made more interesting by being a bit more explicit in discussing possible mechanistic interpretations of the data.

We agree that the conclusions section did not do the results justice, which was also mentioned by reviewer #2. We have rewritten the conclusions completely.

- The Ms adds some new information on the various subcomplexes in the multipass translocon and would be suitable as a short communication.

Minor issues:

- Figs 2a and 2b would benefit from a legend.

We expanded on the figure caption, and we explained the colors by a legend in panels A and B for convenience, which is probably what the reviewer is referring to.

- In the "Data availability" section: "We made use of a previously published atomic models..." (remove "a")

Removed.

- In the first paragraph of the results subheading "TRAP interacts with the BOS complex", in the second-to-last sentence, the first words "Vice versa, however" are awkward. The readability might benefit by being replacing that phrase with something simpler, like "alternatively."

Agreed, 'alternatively' fits better.

- In the third paragraph of the results subheading "TRAP interacts with the BOS complex": "An obvious candidate for the unidentified densities is NOMO, a subunit of the BOS complex (citation), because it was shown..." (add "it")

Corrected.

Reviewer #2:

This study describes additional analysis of cryo-ET data recently published in PMID 36697828. The authors focus on the TRAP complex, implicated in facilitating translocation through the Sec61 channel, and its relationship to the multipass translocon (MPT) subcomplexes known as PAT, GEL, and BOS. The main conclusions drawn by the authors are that TRAP and BOS contact each other, potentially through the NOMO subunit of BOS, and that this interaction may influence the relative positions of the TRAP and BOS complexes.

The advance here is rather modest, and it is difficult to draw any appreciable functional implications of the observations or propose any specific hypotheses related to either TRAP or BOS function. This is evident in the "Conclusions" section, which is not able to articulate what was actually advanced by this study beyond rather superficial and vague statements. The main useful contribution beyond the TRAP-BOS relationship is adding to the census of translocon types in native membranes based on

classification of the particles observed by cryo-ET. If one were to publish this analysis, the following suggestions for improvement (in the order they appear in the paper) should probably be incorporated or addressed.

1. The authors are vague about how they are defining the MPT. For example, in the abstract they talk about PAT associating with the MPT, but as per earlier definitions, PAT is an integral component of the MPT. Perhaps for simplicity, the authors are calling all complexes containing BOS and lacking OSTA the MPT, as this can be readily distinguished in the cryo-ET data? Regardless, the authors need to define their terms and use them consistently.

We agree that a clear definition of terminology adds clarity. We now refer to as the MPT as the fully assembled Sec61-TRAP-BOS-GEL-PAT or $MPT_{TRAP+/PAT+}$ and define the smaller variants as $MPT_{TRAP+/PAT-}$, $MPT_{TRAP-/PAT+}$, and $MPT_{TRAP-/PAT-}$. Consistent with our interpretation in Fig. 6 we now describe which subunits are missing compared to the fully assembled MPT.

2. The authors talk about active and inactive translocons. What does 'active' mean, and how are they defining this? I assume they cannot see substrate, so this cannot be the deciding parameter. Also, it presumably does not reflect the open or closed state of Sec61. Perhaps they are talking about whether the associated ribosome is translating or not? If so, then they should precise and say something like "...translocons associated with translating ribosomes showed... whereas translocons associated with non-translating ribosomes showed..." I realise it is cumbersome, but the lack of precision in how they describe the findings makes the paper very difficult to understand, even by an expert. It is also important to be clear about exactly what metric is used to assign something as translating or not, active or inactive, etc.

We agree that referring to translating and non-translating ribosome-MPT particles is more specific and changed the text accordingly.

3. Still on terminology, how is hibernating defined, and why use this rather than inactive? I think hibernation means something different than inactive, and it is not clear why hibernating ribosomes would associate with translocons anyway.

We clarify hibernation in the revised text: "Translating ribosomes could be assigned to the elongation cycle, while most non-translating ribosomes displayed the factor CCDC124 that is associated with hibernation (Wells et al, 2020)."

Association of inactive ribosomes has also been reported earlier, most notably by the Nicchitta lab, and the functional role of this interaction indeed remains to be clarified further.

4. In the abstract, the conclusion that TRAP associates with inactive translocons is going to be very misleading. There is little doubt that TRAP is part of translocating Sec61 complexes (plenty of past biochemical and structural work to support this), and indeed the authors know this as they talk about the TRAP cradle associating with substrates and so forth.

We re-phrased the abstract focusing on activity of the associated ribosomes, as opposed to biogenesis factors (see also point 2).

5. The persistent use of native or near-native is again not precise or helpful. The fact that microsomes are isolated necessarily means an important aspect of the sample is not native or near-native - there are countless things that can happen during this biochemical process including dissociation of key factors, loss of tRNA due to hydrolysis, and so forth. I therefore object to such terms. What is native is the membrane environment, and the authors should strive to be precise about this. Again, I realise it is cumbersome terminology, but it will also minimise confusion in the future.

We now consistently refer to 'native' in the context of the membrane environment.

6. The authors should note the caveat that the ribosome-nascent chain state might change during fractionation, altering to some degree the correlation between what is seen at the ribosome (i.e., translating or not) and what is seen in the membrane. This is important, because some of the claims are at odds with earlier conclusions. For example, work by Smalinskaite et al. and Sundaram et al. indicates that the MPT is assembled in response to substrate, and the data supporting this seems fairly clear. So one would not expect to see the MPT associated with non-translating ribosomes, so observing this population warrants some explanation. Being explicit about the caveats will be helpful in providing some possible resolutions to this difference in observations.

We clarify our interpretation extensively in the revised manuscript. Our data are consistent with the assembly of the MPT as a response to a substrate. We suggest that the variability we observe is rather due to partial dissociation of PAT after fulfilling its function, while BOS remains associated. We clarify that the failure to capture complexes of non-translating ribosomes with MPT upon isolation is likely a solubilization effect, which would be analogous to the 'secretory translocon'.

7. Introduction, paragraph 1: the sentence starting "Sec61 assembles with..." is redundant with the previous sentence and can be deleted.

Agreed. We deleted the sentence.

8. Introduction, paragraph 2: the GET pathway references are not appropriately chosen. The initial discovery that post-translational targeting of TA proteins is mediated by GET3 was reported by Stefanovic & Hegde, 2007, Cell. The demonstration that the membrane subunits directly mediate TA protein insertion was first shown in reconstitution experiments by Mariappan et al., 2011, Nature and Wang et al., 2011, Mol. Cell. The Schuldiner paper could not really draw conclusions about insertion, only that GET1/2 are a receptor for GET3. The role of GET as an insertase was shown by Wang et al., 2014, Nature.

We thank the reviewer for the explanation and included the references.

9. Introduction, paragraph 2, last sentence: affinity tagged TMCO1 was only part of the strategy; the strategy used by Smalinskaite et al. and Sundaran et al. was affinity tagged defined membrane

protein insertion substrates. Probably best to split into two sentences and clarify what was done in which study.

As suggested, we split sentence as: “Biochemical, mass-spectrometry and cryo-electron microscopy (cryo-EM) studies of affinity-tagged TMCO1 isolates revealed that TMCO1 is part of a ribosome-bound translocon that is distinct from its OSTA-containing counterpart and engages with multipass-transmembrane substrates (McGilvray et al., 2020). More detailed analysis utilizing specific substrates detailed the components and function of what is herein referred to as the multipass translocon (MPT) (Smalinskaitė et al, 2022; Sundaram et al, 2022).”

10. Introduction, paragraph 3, line 2: "complexes" should be "complex".

Done.

11. Introduction, paragraph 3: The GEL complex was not defined by McGilvray et al., 2020, as they did not know about OPTI. The other references cited are appropriate (Lewis, Smalinskaite, Sundaram).

Agreed. We removed the reference in this context.

12. Introduction, paragraph 3: The finding that CCDC47 impedes Sec61 opening was shown in Smalinskaite et al., 2022, not in either Chitwood or Meacock.

We changed the citations accordingly.

13. Introduction, paragraph 3: Meacock cannot realistically be cited for the PAT complex. They of course did pioneering work in detecting a crosslink near membrane protein insertion intermediates, but did not know what that crosslink was, what its function was, or that it had CCDC47 as a partner.

We removed Meacock et al from the references.

14. Introduction, paragraph 3: McGilvray didn't have any functional data about what the multipass translocon does. The role in TMH insertion was shown later in the 2022 Nature papers.

We clarified the contribution of McGilvray et al in paragraph 2 and referenced the two nature papers in this instance.

15. Introduction, paragraph 3, last sentence: this is potentially misleading. Sundaram certainly examined the activity-dependent compositional variation, and to a lesser extent, so did Smalinskaite. Both these studies clearly established substrate-triggered MPT assembly and disassembly. I think the authors mean to emphasise the native membrane part of the sentence, so it should be re-phrased to focus on this.

To avoid possible misunderstanding, we re-formulated the sentence as: “While biochemical analysis and cryo-EM studies revealed the organization of the detergent-solubilized MPT bound to substrates, its compositional variation in the native membrane and the correlation with ribosomal activity remains to be explored.”

16. Pg. 5, paragraph 1: the conclusion that TRAP is recruited to inactive translocons is incompatible with the findings that TRAP in fact is needed during translocation (e.g., to initiate the process by

facilitating signal sequence insertion; Fons et al., 2003, JCB), and also crosslinks to translocating polypeptides (e.g., Gorlich et al., 1992, Nature). There is also earlier structural data on the TRAP complex and this too suggests it associates with translating and translocating translocons. The authors need to both be careful about terminology and also discuss the relationship of this work to earlier studies.

We agree that the formulation could cause unnecessary confusion. Clarification that only translational activity can be determined in our data (point 2) is indeed helpful to avoid misunderstanding.

We also re-formulated the relevant section in the 'Results and Discussion' section (see also point 6).

17. Using DTT as a surrogate for modulating translation is just an awful idea as DTT will have many, many other effects. I think this needs to be acknowledged. Why they don't use a translation initiation inhibitor is unclear, but regardless, they should acknowledge the pleiotropic effects of DTT. Also, mention the time and concentration of treatment in the Results so one does not have to hunt down this key piece of information.

We re-formulated the motivation for inducing UPR. In point 6, reviewer #2 raises the point that one would not expect to find MPTs bound to non-translating ribosomes (in principle, the same holds for OST in the case of the secretory translocon). The UPR is a physiologically important condition that induces accumulation of non-translating ribosomes on the ER surface. While we acknowledge that the many aspects of the UPR in general and DTT in particular are not ideal to study effects of translation, the intent was not to study relatively fast responses to the translational state.

In the revised manuscript we clarified that DTT treatment has effects on the cell beyond reduction of translation and we added the relevant details of DTT treatment to the results section (10 mM DTT for 2 h).

18. The authors might wish to discuss and incorporate into their thinking the new preprint by Lewis and Hegde (BioRxiv). This shows that TRAP-alpha can remain associated whereas the membrane regions of the other subunits are disordered due to the luminal domain being shifted by the BOS complex.

We extensively discuss this preprint that appeared after submission of initial manuscript in the revised manuscript.

19. Pg. 5, bottom: The "substrate-dependent recruitment behavior of PAT during translocation" was also shown in Sundaram et al., 2022.

Agreed. We added the reference.

20. Pg. 6, top: McGilvray didn't even know about Asterix, so it is unclear why this is cited for function of the PAT complex. The appropriate citation is Chitwood & Hegde, 2020, Nature.

Agreed. We exchanged the reference.

21. Pg. 6, top: The sentence "we speculate that PAT associates..." is not really speculation. This was shown directly in Smalinskaite et al., 2022, but mutating the hydrophilic residues of the substrate TMH and analyzing the consequences for MPT assembly.

We re-formulated the respective section as it was partially redundant.

22. The model by McGilvray (PDB: 6W6L) lacks the TRAP complex, but the authors should probably have a careful look at the maps as I believe it does contain the TRAP complex as evident by its prominent luminal domain.

Indeed, EMD-21435 contains some density corresponding to TRAP when filtered to low resolution. However, the positioning is distinct from that observed in our structure and that by Lewis et al. We changed the relevant section in our manuscript to "... which displays the TRAP complex in an unique position distant from BOS and Sec61."

23. Pg. 6 bottom: mass spectrometry probably cannot distinguish between NOMO1, 2 and 3, so perhaps re-phrase that sentence. The supposed enrichment of NOMO1 and 3 is probably just an annotation issue.

We aligned the three sequences and indeed the high sequence identity between the three paralogs makes accurate quantification challenging. Hence, we re-phrased as advised to: "Large stretches of the three NOMO paralogs in the human genome are identical, which makes it challenging to specify the relative expression levels of specific paralogs by mass spectrometry."

24. Pg. 7, bottom: I cannot see how the word "comprehensive" is justified in the Conclusions section. The overall number of particles is really too small to do thorough classification and it is worth being a bit less bombastic and a bit more realistic here.

We have re-written the conclusion, including the removal of 'comprehensive'.

Reviewer #3:

Review of manuscript of Gemmer et al. entitled: 'Exploring the molecular composition of the ER multipass translocon in its native membrane environment'

This manuscript describes the cryo-electron tomography structure characterization of co-translational ribosome/Sec61 complexes together with the translocon associated protein complex (TRAP) and components of the recently described multipass translocon and the intramembrane chaperone PAT. The work builds upon previous recent work from the same group (Gemmer et al, 2022) and recent body of work describing functions and structures of different Sec61 complexes involved in insertion of multipass membrane proteins. Collectively, the latter complexes are termed the multipass translocon, which comprises of three subcomplexes (GEL, PAT and BOS complexes).

The work presented here comes from further analysis and classification of the compositionally diverse multipass translocons, using data collected in the earlier Gemmer et al. study. Specifically, the authors expand the analysis of Sec61 recruitment of TRAP versus components of the multipass translocon on Sec61 and demonstrate that TRAP appears to preferentially associate with Sec61 bound to inactive ribosomes, whereas multipass translocon associated with actively translating ribosomes.

A new finding is identification of an unknown density in Sec61 complexes containing TRAP and the BOS subcomplex of the multipass translocon. The authors speculate that this density would represent membrane-proximal subunit of the NOMO subunits of the BOS complex, although the low resolution of the experimental density precludes unambiguous assignment. A model built with Colabfold fits well to the observed density, providing support to the proposed identity of NOMO in this location bound to the ER luminal domains of TRAP.

Major points:

1. The resolution of the reconstruction is quite modest, which prevents unambiguous assignment of the putative NOMO Ig-like domains. I realize that testing the Colabfold-derived model of NOMO-NCLN remains a future task, it would be important to indicate what the main proposed contacts in the Colabfold model are.

We specified the interaction residues in the manuscript: "In the model, the NCLN residues 420-458 and 478-486 interact with the NOMO1 residues 969-978, 1027-1033, 1049-59, 1097-1099 and 1137-1140 located in the last two C-terminal domains."

Minor points:

1. The authors suggest that TRAP would preferentially associate with inactive translocons based on the finding that TRAP is preferentially enriched with inactive ribosomes in the cryo-ET analysis. Is it believed that TRAP would associate with Sec61 already before ribosome docking?

We clarified in the conclusions that we suggest that TRAP-Sec61 translocons preferentially interact with ribosomes early during synthesis. Whether TRAP associates with Sec61 in the absence of the ribosome is unclear at this point. The relatively small interaction surface of TRAP and Sec61 suggests that this is not very likely, though. We note that work from the Nicchitta lab suggests that translation can also be initiated on ER-bound ribosomes, which may explain the functional role of inactive ribosomes on the ER surface and their bound translocons (see also point 3 of reviewer #2).

2. Page 7. The authors state that "our results strongly suggests that the luminal domain of NOMO mediates association of TRAP and NCLN in the ER lumen." Does the current placement of the Colabfold-derived model of NOMO-NCLN suggest what the interaction site on TRAP would be? It appears that this interface would involve residues of TRAP α and TRAP β . Could the authors indicate

the TRAP putative interaction surface and highlight some putative interacting residues of the TRAP luminal domains? This would ideally take place as a new additional figure panel.

We added a close-up view to Fig. 5B showing the interaction site between NOMO and TRAP and indicated the interacting residues in the figure caption: NOMO 1068-1072, 1117-1120 with TRAP α 90-94, 191-195, and NOMO 1084-1089 with TRAP β 51-53, 83-86

May 15, 2024

RE: Life Science Alliance Manuscript #LSA-2023-02496-TR

Prof. Friedrich Förster
Utrecht University
Bijvoet Centre for Biomolecular Research
Universiteitsweg 99
Utrecht 3584 CG
Netherlands

Dear Dr. Förster,

Thank you for submitting your revised manuscript entitled "Exploring the molecular composition of the multipass translocon in its native membrane environment". We would be happy to publish your paper in Life Science Alliance pending final revisions necessary to meet our formatting guidelines.

- please be sure that the authorship listing and order is correct
- please upload all figure files as individual ones, including the supplementary figure files; all figure legends should only appear in the main manuscript file
- please add ORCID ID for the corresponding (and secondary corresponding) author--you should have received instructions on how to do so
- please add a Summary Blurb/Alternate Abstract to our system
- please add the Twitter handle of your host institute/organization, your own, or one of the authors to our system.
- please incorporate any points from the Conclusion section into the Results and Discussion
- please add your main and supplementary figure legends to the main manuscript text after the references section
- remove figures from the manuscript file and upload them separately
- please add callouts for Figures 3A-B; 4A-B; S2A-B; S3A-B and S4A-E to your main manuscript text
- in the Materials and Methods section, reference to (Gemmer et al., 2023) is fine, however details should be provided here as well to avoid readers having to cross-reference for methods used.

A. FINAL FILES:

B. MANUSCRIPT ORGANIZATION AND FORMATTING:

Sincerely,

May 23, 2024

RE: Life Science Alliance Manuscript #LSA-2023-02496-TRR

Prof. Friedrich Förster
Utrecht University
Bijvoet Centre for Biomolecular Research
Universiteitslaan 99
Utrecht 3584 CH
Netherlands

Dear Dr. Förster,

Thank you for submitting your Research Article entitled "Exploring the molecular composition of the multipass translocon in its native membrane environment". It is a pleasure to let you know that your manuscript is now accepted for publication in Life Science Alliance. Congratulations on this interesting work.

DISTRIBUTION OF MATERIALS:

Again, congratulations on a very nice paper. I hope you found the review process to be constructive and are pleased with how the manuscript was handled editorially. We look forward to future exciting submissions from your lab.

Sincerely,
